# Adaptive divergence and post-zygotic barriers to gene flow between sympatric populations of a herbivorous mite

Ernesto Villacis-Perez [1,2,5✉], Simon Snoeck [2,4,5], Andre H. Kurlovs[2], Richard M. Clark [3], Johannes A. J. Breeuwer[1✉] & Thomas Van Leeuwen[2✉]

Plant-herbivore interactions promote the generation and maintenance of both plant and herbivore biodiversity. The antagonistic interactions between plants and herbivores lead to host race formation: the evolution of herbivore types specializing on different plant species, with restricted gene flow between them. Understanding how ecological specialization promotes host race formation usually depends on artificial approaches, using laboratory experiments on populations associated with agricultural crops. However, evidence on how host races are formed and maintained in a natural setting remains scarce. Here, we take a multidisciplinary approach to understand whether populations of the generalist spider mite *Tetranychus urticae* form host races in nature. We demonstrate that a host race co-occurs among generalist conspecifics in the dune ecosystem of The Netherlands. Extensive field sampling and genotyping of individuals over three consecutive years showed a clear pattern of host associations. Genome-wide differences between the host race and generalist conspecifics were found using a dense set of SNPs on field-derived iso-female lines and previously sequenced genomes of *T. urticae*. Hybridization between lines of the host race and sympatric generalist lines is restricted by post-zygotic breakdown, and selection negatively impacts the survival of generalists on the native host of the host race. Our description of a host race among conspecifics with a larger diet breadth shows how ecological and reproductive isolation aid in maintaining intra-specific variation in sympatry, despite the opportunity for homogenization through gene flow. Our findings highlight the importance of explicitly considering the spatial and temporal scale on which plant-herbivore interactions occur in order to identify herbivore populations associated with different plant species in nature. This system can be used to study the underlying genetic architecture and mechanisms that facilitate the use of a large range of host plant taxa by extreme generalist herbivores. In addition, it offers the chance to investigate the prevalence and mechanisms of ecological specialization in nature.

[1] Institute for Biodiversity and Ecosystem Dynamics (IBED), University of Amsterdam, Amsterdam, Netherlands. [2] Department of Plants and Crops, Faculty of Bioscience Engineering, Ghent University, Gent, Belgium. [3] School of Biological Sciences and Henry Eyring Center for Cell and Genome Science, University of Utah, Salt Lake City, UT, USA. [4] Present address: Department of Biology, University of Washington, Seattle, USA. [5] These authors contributed equally: Ernesto Villacis-Perez, Simon Snoeck. ✉email: e.a.villacisperez@uva.nl; j.a.j.breeuwer@uva.nl; Thomas.VanLeeuwen@UGent.be

For herbivorous arthropods, the transition to a plant-eating lifestyle has contributed to their immense diversity. Herbivorous arthropods are highly speciose[1,2], and a substantial proportion of speciation events in insect herbivores can be attributed to shifts towards novel host plant taxa[3]. Plants and herbivores exert strong selective pressures on each other. These antagonistic interactions may promote local adaptation and potentially host race formation. Following Drès and Mallet[4], host races are defined as populations that (i) are associated with their hosts across spatial and temporal scales; (ii) show some extent of genetic differentiation from sympatric conspecifics; and (iii) show incomplete reproductive isolation from sympatric conspecifics. In addition, host races may differ from conspecifics in traits associated with host plant adaptation, but this criterion is not an absolute requirement for host race formation[4–6]. Host races have been identified in maple- and willow-infesting *Neochlamisus bebbianae* beetles[6,7], in *Rhagoletis* flies associated with apple or hawthorn[8–10], in the pea aphid *Acyrthosiphon pisum*[11–13], in populations of the peach-potato aphid *Myzus persicae* infesting tobacco[14,15], as well as in several other herbivores[3,4]. In nature, the composition of vegetation varies across spatial and temporal scales, and the availability of certain hosts can be restricted across clines in the landscape[16–18]. Herbivores with large geographic ranges may encounter a myriad of plant taxa, some more heavily defended against herbivory than others. This results in a mosaic of locally available hosts, in which herbivore population structure can arise due to spatially and temporally varying selection, resulting in local adaptation and extinction events[19,20]. Local variation in the strength of host selection can contribute to the evolution of host races[4,6,21].

Whether generalist herbivores adapt and diverge via the interaction with their host plant is not clear. Most herbivore arthropods are specialists; they only accept species within a single plant family or even one or a few closely related species[22–24]. Specialist herbivores exploit their hosts by: (i) behavioural avoidance of physical or chemical plant defences, (ii) decreasing the impact of chemical defences through various detoxification enzymatic pathways that are deployed upon exposure to plant-derived defensive metabolites, and (iii) the active transport of these metabolites towards excretion or sequestration[25–27]. Only a few examples exist of true generalist herbivores occurring on host plant species across families[23]. Generalists are thought to feed on a large range of plant taxa either by plasticity in characters that determine host range, or by forming cryptic complexes of host-adapted genotypes, like in the whitefly *Bemisia tabaci*[28]. The ecological interactions between generalists and the plethora of hosts they use through time in the field are not well understood, and knowledge of the extent to which cryptic complexes of host races within a taxon contribute to these interactions remains scarce[29].

The two-spotted spider mite (*Tetranychus urticae*; Acari: Tetranychidae) has an extremely large host range spanning over a 1000 plant species, distributed over 120 families, some of which are important crops[30,31]. Two colour morphs of *T. urticae* occur across its cosmopolitan distribution, the green and the red morphs[32]. The role of host adaptation in promoting population structure and reproductive isolation in *T. urticae* in nature is not clear, as it is uncertain whether it is a complex of host races or whether its breadth in host range reflects variation in host preference[33–35] or plasticity in its molecular toolkit[36–47].

Previous research on the subject is conflicting. Firstly, population differentiation based on genetic markers has rarely been found to correlate with colonized host plant species. Instead, genetic differentiation has been found to decrease when populations are closer to each other and when population densities are large[36,48–51]. This is likely due to sexual reproduction homogenizing the genetic pool when populations associated with different hosts encounter each other. Still, fine-scale genotyping of individuals has shown that population structure exists[40,41,43,49,52]. These findings suggest that reproductive barriers between individuals may maintain population structure, even when gene flow seems likely due to physical proximity. Moreover, population differentiation based on genetic markers has rarely been coupled with assays quantifying reproductive compatibility[39]. Research quantifying hybrid inviability in this species is largely confounded, because the infection status of experimental populations with strains of common mite endosymbionts, which cause similar patterns of reproductive incompatibility, is often not reported[53–57]. Lastly, experimental assays suggest a genetic basis for increased fitness traits on novel host species, yet most studies that have analysed whether field-collected mite populations are adapted to their host plants have not been coupled with molecular genetic data[42], but see refs. [47,58].

Here, we aimed to quantify the extent of genetic differentiation and reproductive isolation between co-occurring populations of *T. urticae* associated with different plant species in nature. To achieve this, we conducted a holistic analysis that integrates four complementary approaches: (i) extensive field sampling and genetic screening of individuals on multiple host species across a single ecosystem over three consecutive years; (ii) screening for genome-wide and localized patterns of sequence differentiation using field-derived iso-female lines and previously sequenced populations of *T. urticae*; (iii) quantification of the extent of reproductive isolation between field-derived lines; and (iv) quantification of fitness traits in field-derived lines to test for host plant adaptation. We present clear evidence for the occurrence of a specialist *T. urticae* host race, which is isolated from conspecifics by a post-zygotic barrier to gene flow, and that is genetically differentiated from sympatric generalist conspecifics that co-occur with the host race at small spatial scales.

## Results and discussion

**Consistent associations between sympatric mite cytotypes and their host species in nature.** To determine if sympatric *T. urticae* populations showed genetic differentiation based on host species across spatial and temporal scales, we extensively sampled and genotyped individual mites from several co-occurring plant species over the course of three years. Specifically, we looked for consistent associations between spider mite *cytotypes* (inferred from mitochondrial *cytochrome oxidase I* [*CO1*] haplotype groups) and several host plant species in two nature reserves of The Netherlands—the dunes near Castricum and the dunes near Meijendel in the summers of 2015, 2016 and 2017.

We determined 598 bp-long *CO1* sequences of 1023 individuals collected over the three years of field sampling from 48 locations (30 in Castricum and 18 in Meijendel). All sequences closely matched GenBank sequences previously annotated and validated as *T. urticae*[59]. The 1023 sequences represented 156 different haplotypes. A maximum-likelihood phylogenetic analysis grouped these haplotypes into six clades (hereafter called cytotypes, followed by a number, i.e., C1 through C6) varying in their bootstrap support (Supplementary Fig. 1). While cytotypes C2 and C3 were found on every host sampled, C1 was restricted to honeysuckle (*Lonicera peryclimenum*), and these association patterns were consistent across the three years of sampling. Cytotypes C4, C5, C6 were rare in the field and were not documented every year; thus, we focused subsequent analyses on C1, C2 and C3, to which the majority of individuals (997 mites) were assigned; 20% of the 997 individuals were assigned to C1, 57% to C2, and 23% to C3 (Fig. 1). The restriction of cytotype C1 to honeysuckle did not seem to be an artefact of our sampling design. Even in locations where mites from any cytotype had an

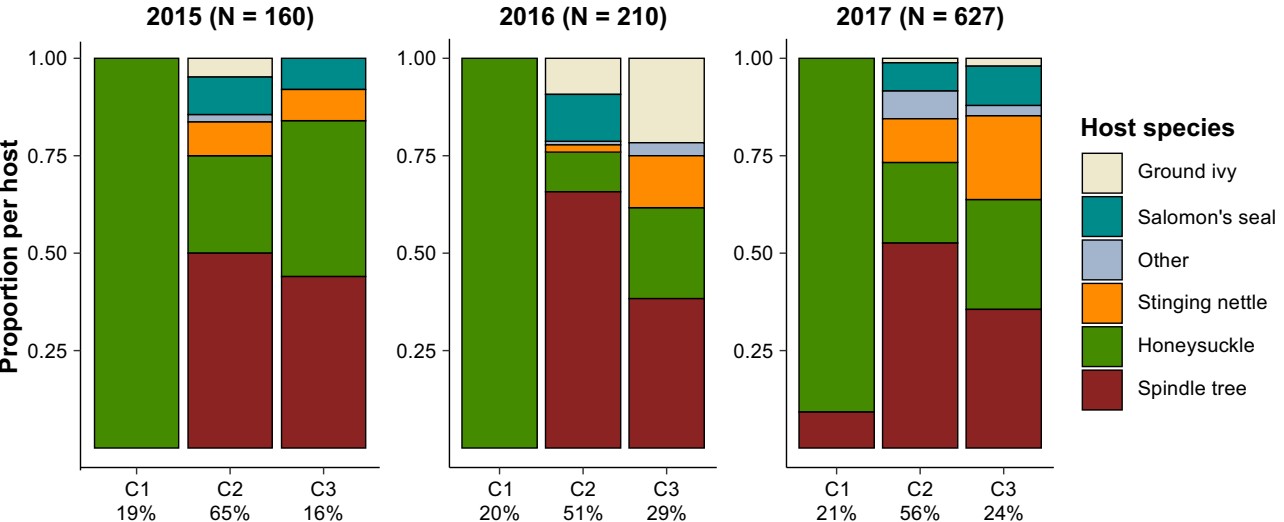

**Fig. 1 Associations between *Tetranychus urticae* cytotypes and host plant species in the dune ecosystem of The Netherlands.** Host range of the three most abundant *Tetranychus urticae* CO1-based cytotypes (C1, C2 and C3) identified in 2015, 2016 and 2017 (left, middle, and right, respectively). Percentages on the *x*-axis represent the number of individuals belonging to a cytotype over the total number of individuals (*N*) per year. Proportions on the *y*-axes represent the number of individuals of a cytotype found on a particular host over the total number of individuals belonging to that cytotype. Host species (legend, far right) are ground ivy (*Glechoma hederacea*), wild honeysuckle (*Lonicera periclymenum*), Salomon's seal (*Polygonatum multiflorum*), spindle tree (*Euonymus europaeus*), and stinging nettle (*Urtica sp.*). 'Other' includes blackberry, stalk rose, elderberry, and at least three different host-species that were not taxonomically identified.

equal chance of colonizing either of two co-occurring widespread hosts, honeysuckle or spindle tree (*Euonymus europaeus*), more than 97% of C1 individuals were found on honeysuckle. The proportion of C1 individuals found on spindle tree was not significantly different from zero ($t_{(21)} = 1.28$, $p = 0.21$), and a high proportion of C2 and C3 individuals were also found on honeysuckle (Supplementary Fig. 2). *T. urticae* mostly disperses passively with the wind and occasionally by actively walking to close-by plants[60,61]. We only once (2017) found a few C1 individuals on spindle tree shoots growing underneath a large honeysuckle patch which was highly infested with C1 individuals; this large C1 population likely spilled over but did not establish on spindle tree.

The spatial scale at which the processes that lead to mosaics of genetic variation in herbivore taxa occur largely depends on the local and regional availability of suitable hosts[20,62,63]. For example, the maintenance of genetically structured populations of the whitefly *Bemisia tabaci* depends on the seasonality of host availability, which influences interpopulation gene flow by temporally isolating *B. tabaci* populations associated with hosts with different phenologies[64,65]. Similarly, variation in the availability and phenology of local hosts across the United States and Mexico promotes the maintenance of apple- and hawthorn-associated host races of *Rhagoletis* flies[10]. In contrast, *T. urticae* genetic population structure has not been previously shown to depend on host species, regardless of the spatial scale at which these hosts occurs[38,39,44]. Moreover, prior studies have not investigated the temporal persistence of spider mite genotypes in the field. In this study, extensive sampling and genotyping of individuals within and between populations on a small spatial scale, across consecutive years, revealed the spatial scale that promotes spider mite population structure across time in the field.

**Genome-wide divergence of honeysuckle-restricted lines relative to sympatric and non-sympatric conspecifics.** To confirm and further assess the extent of genetic divergence between host-

associated populations, we also asked if the *CO1*-based cytotypes were consistent with full mitochondrial genomes, as well as with patterns of nuclear variation. To do this, we established 26 iso-female lines from cytotypes C1, C2 and C3 in the laboratory, sequenced the lines using the Illumina method, and predicted single nucleotide polymorphisms (SNPs) for the 90 Mb genome of *T. urticae*. With a high-quality subset of SNPs, we constructed mitochondrial and nuclear phylogenies. We found that the phylogeny based on the whole mitochondrial genome was congruent with the phylogeny based on *CO1* haplotypes (Supplementary Fig. 3). The nuclear phylogeny also split the 26 lines into three clades, hereafter referred to as *nucleotypes* N1, N2 and N3. While each of the three nucleotypes had >95% bootstrap support, the N1 clade grouped away from N2 and N3, with the latter in a well-supported clade (bootstrap support 100%) (Fig. 2). The 26 lines were subsequently labelled based on their *genotype*, this is, their cytotype and nucleotype together; e.g. lines belonging to cytotype C1 and to nucleotype N1 belong to genotype group C1N1 (Supplementary Table 1).

To further explore genetic relationships, we performed principal component analyses (PCA) with the nuclear data of the 26 iso-female lines, and included nuclear data of 30 previously sequenced lines of *T. urticae* green forms, two lines of *T. urticae* red forms (MR-VL and TuSB9), and two sister species (*T. turkestani* and *T. kanzawai*) (Fig. 3a). As expected, *T. kanzawai*, for which no viable progeny from interspecific crosses with *T. urticae* have been reported[66], was separate from all *T. urticae* lines along PC1. Previously, viable but infertile F1 progeny have been reported in crosses between *T. urticae* and *T. turkestani*[67], which clustered separately from all *T. urticae* lines along PC2. All green forms of *T. urticae*, including the 26 iso-female lines derived in this study, formed a tight cluster distinct from *T. urticae* red forms, for which varying levels of incompatibility have been reported[56,68]. A PCA restricted to nuclear data for the 56 *T. urticae* green forms revealed that lines of nucleotype N1 clustered together and away from all other green strains along PC1; along PC2, all the lines from our study sites clustered separately from lines reported previously from other geographic locations, as

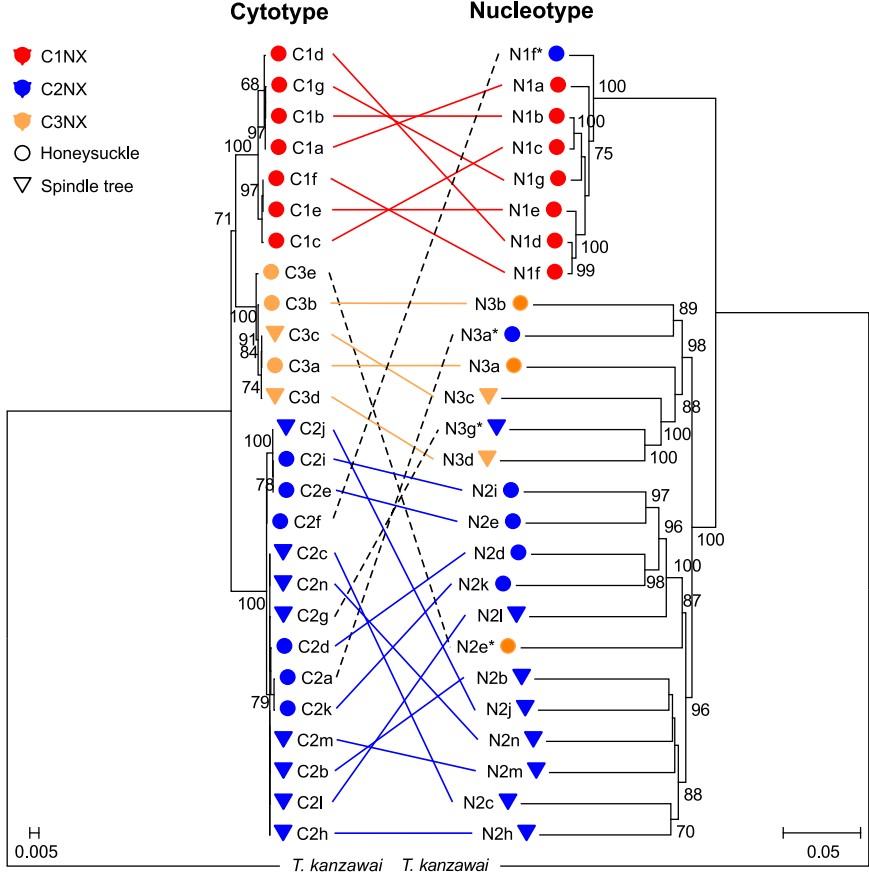

**Fig. 2 Mitochondrial and nuclear genome phylogenies of field-derived *Tetranychus urticae* iso-female lines.** At left is a maximum likelihood analysis based on the optimal partitioning scheme of 13 protein-coding mitochondrial genes. At right is a neighbour-joining analysis using whole nuclear genomes. In both analyses, *Tetranychus kanzawai* is included as an outgroup. The scale bar of the mitochondrial phylogeny represents the nucleotide substitutions per site, the scale bar of the nuclear genome phylogeny represents substitutions per nuclear SNP position. Mitochondrial and nuclear phylogenies for the same field-derived lines are labelled and connected by lines, with colours according to their cytotype and shapes according to their native hosts. Cyto-nuclear hybrids are connected with dotted black lines and have asterisks next to their nuclear name. Only bootstraps values ≥65 are shown.

expected for local population differentiation (Fig. 3b). The PCA on nuclear data are therefore consistent with species and population delineations (red versus green forms) reported previously. The PC analyses also revealed no apparent separation of N2 and N3 by host plant species, although the N2 and N3 lines did appear to have a modest genetic separation based on geography (e.g. N2 lines grouped closer to each other based on sampling site, while all members of N3 were collected in Meijendel (Supplementary Fig. 4E).

We also quantified the extent of genetic differentiation between field-derived lines by calculating genome-wide Fst values. Genetic differences between N1 and the other two nucleotypes were substantially higher than between N2 and N3 ($\text{Fst}_{\text{N1}-\text{N2}} = 0.46$, $\text{Fst}_{\text{N1}-\text{N3}} = 0.54$, $\text{Fst}_{\text{N2}-\text{N3}} = 0.07$; Fig. 3c). To understand whether localized genomic regions of unusually high or low genetic differentiation were apparent, we quantified Fst in sliding windows across the genome among the nucleotype groups. We found that divergence between N1 and N2/N3 was approximately uniform along each of the three *T. urticae* chromosomes (Fig. 3d). Given their co-occurrence at field sites, the genome-wide uniformity of divergence of N1 from the other nucleotypes as assessed with Fst was unexpected. A potential explanation for this finding was revealed by an analysis of genetic variation within nucleotypes. As our study used iso-female lines (expansions from single diploid females), residual genetic variation was anticipated in each line. Consistent with this expectation, segregation of genetic variants was observed for large genomic regions in the N2

and N3 iso-female lines. In striking contrast, each of the N1 lines was inbred, or nearly so, as assessed with the dense, genome-wide SNP data (Supplementary Fig. 5). This pattern can only be explained if the females used for establishing the N1 lines were isogenic (or largely so) at the time of collection from the field. This observation, coupled with the finding that the N1 lines are predominantly identical by allelic state genome-wide (Supplementary Fig. 6), revealed that our N1 nucleotype constituted a highly inbred population across our study locations in nature; this is expected to elevate Fst values in population comparisons[69]. The N1 lines were not uniformly inbred, however, as revealed by the phylogenetic and PC analyses (Figs. 2 and 3b), and a few small genomic intervals were segregating in a few N1 lines (Supplementary Fig, 5A). Moreover, while twenty-two out of the twenty-six lines formed analogous clades with respect to both the cytotype and the nucleotype, (i.e., N1, N2 and N3 were consistent with C1, C2 and C3, respectively), there were four exceptions (hereafter referred to as *cyto-nuclear hybrids*, Fig. 2). One of these involved a C2 cytotype and an N1 nucleotype, line C2N1f, for which the nuclear genome was most distinct within the N1 nucleotype group (Figs. 2 and 3b), potentially reflecting introgression in nature.

Together, these data suggest that gene flow is prevalent within and between the N2 and N3 nucleotypes, while the N1 nucleotype co-occurring at minute spatial scales exists across collection sites largely as a genetic clone. The level of homozygosity of N1 resembles those expected of inbreeding in small, isolated

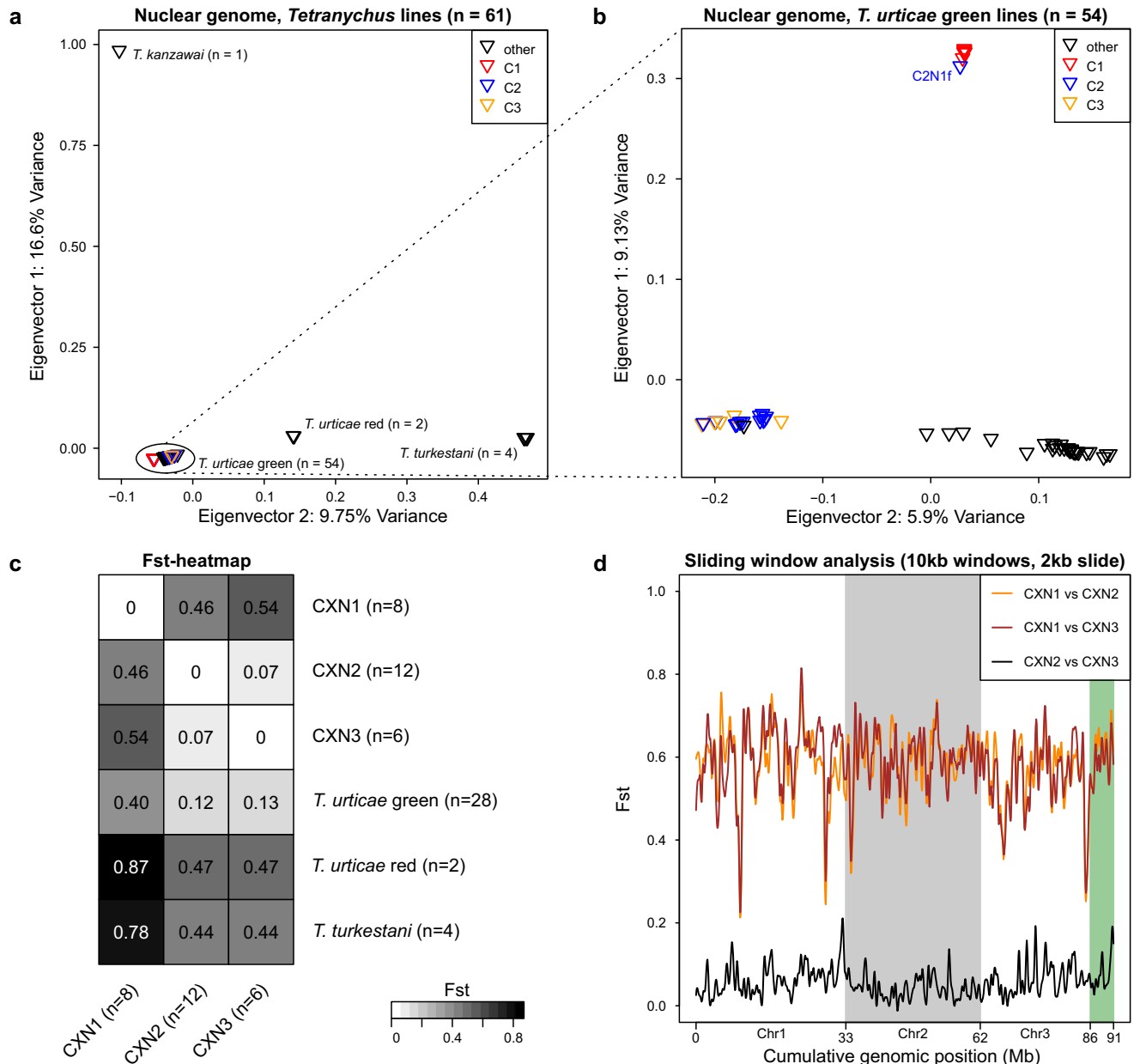

**Fig. 3 Extent of genetic divergence between _Tetranychus species_ and among _Tetranychus urticae_ lines.** Genetic differentiation between 26 iso-female _Tetranychus urticae_ (green) lines (this study), 30 additional _T. urticae_ lines (28 green and 2 red lines, denoted as 'other' and green or red as indicated) and two other _Tetranychus_ species (_T. turkestani_ and _T. kanzawai_) as assessed by principal component analyses (**a**, **b**), as well as by **c** pairwise genome-wide Fst calculations (the extent of shading reflects Fst levels as indicated, bottom right). For **b** genetic differentiation assessed by a principal component analysis, but limited to _T. urticae_ green lines (see dashed lines connecting to **a**), is shown. **d** Genetic differentiation among the 26 iso-female green lines (this study) as assessed in a sliding window Fst analysis across the _T. urticae_ genome (the three scaffolded chromosomes are as indicated with alternative shading; unplaced and concatenated scaffolds are shaded green, far right).

populations[70]. Whether this is a general pattern of host race formation in this system is unclear. In yeast, the early stages in the dynamics of experimental adaptation to a novel environment are accompanied by a loss of genetic diversity[71]. Haplodiploid taxa, such as tetranychid mites and parasitoid hymenopterans, may exist as inbred populations in nature[72,73]. Haplodiploids are expected to suffer less from inbreeding than taxa with other sex determinations systems due to a lower genetic load, as deleterious recessive mutations are purged via exposure in haploid males[72,74].

**Reproductive isolation between sympatric mite genotypes and genetics of cyto-nuclear hybrids.** To quantify the extent of

reproductive isolation in this system, we assessed the extent of reproductive compatibility between field-derived lines. We performed pairwise reciprocal crosses between individuals of several lines of C1N1, C2N2 and C3N3, and assayed how traits associated with hybrid breakdown, such as percentage of sterile F1 females, F1 sex ratio, and F2 egg mortality, compared between hybrids and parental lines (Supplementary Tables 2 and 3). We found consistent evidence of F1 and F2 hybrid breakdown between members of C1N1 and the other two genotype groups. In contrast, hybrid breakdown was not as apparent in crosses between C2N2 and C3N3 (Fig. 4, Supplementary Tables 2 and 3). These patterns could potentially be confounded by the presence of symbiotic

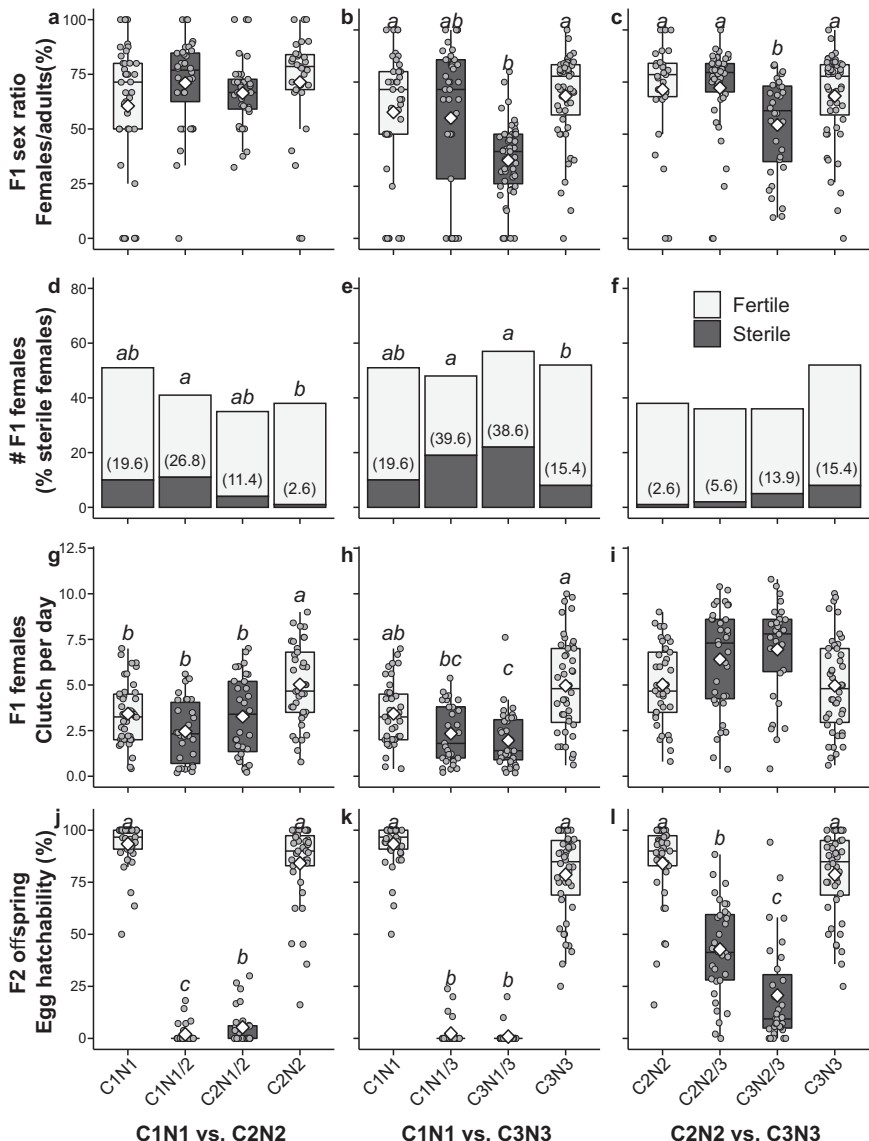

**Fig. 4 Reproductive compatibility of field-derived *Tetranychus urticae* lines.** Reciprocal crosses were performed between different iso-female lines belonging to genotype groups C1N1, C2N2 and C3N3. Fitness traits measured in individuals across two filial generations are presented on the *y*-axes of each row. F1 adult sex ratio, F1 female clutch per day, and F2 egg hatchability are presented in boxplots showing each data point (grey dots), the median (black line within the interquartile range box), the mean (white diamond) and the bottom 25% and the top 25% of the data values (whiskers); reciprocal ♀ × ♂ crosses are presented in black boxes and their respective controls in light grey boxes. F1 female sterility is presented as the number of females (fecund females in light grey and sterile females in dark grey) with the percentage of sterile females in parenthesis. Letters above bars and boxes represent significant differences within each panel. Sample sizes, chi-square and F-statistics, and *p*-values are specified in Supplementary Tables 2 and 3.

intracellular bacteria common to mite and insects, which can cause reproductive incompatibility between *Tetranychus* mites e.g. refs. [57,75,76]. To assess the presence of known incompatibility-inducing bacteria (*Wolbachia*, *Cardinium* and *Spiroplasma*) we sequenced bacterial 16S rDNA of our lines. These bacterial taxa were not found in any of the laboratory lines, and neither were any other bacterial sequences correlated to a cytotype or nucleotype (Supplementary Fig. 7).

Interline crosses produced F1 offspring with slightly altered sex ratios compared to controls, but daughters were always present, demonstrating fertilization in all interline crosses (Fig. 4, Supplementary Table 2). The asymmetrical pattern of distorted sex ratios in the F1 can be explained by hybridization revealing recessive maladaptive combinations between the cytoplasmatic background and the nuclear-encoded mitochondrial genes in hybrids, as observed in haplodiploid taxa[75,77–79]. A higher

percentage of sterile F1 hybrid female virgins occurred when C1N1 was crossed to C2N2 or C3N3 compared to controls; non-sterile F1 virgins from these crosses also laid significantly fewer F2 eggs than controls. In striking contrast, both reciprocal C2N2 vs. C3 N3 F1 hybrids showed no higher proportion of sterile females, and even a non-significant trend of larger egg clutch sizes compared to controls (Supplementary Table 3), which resembles the outbreeding effects of heterosis found among compatible loci related to female reproduction[80]. F1 males were not considered in the analysis because they inherit all the genetic material from their mothers and are therefore not hybrids.

Hybrid breakdown between C1N1 and either C2N2 or C3N3 was strongest in F2 male offspring, which are recombinant haploid individuals. Compared to controls, fewer F2 hybrid eggs hatched (<8% vs. >90%), with the embryonic development of most eggs stopping before the stage of red-eye formation, which is

the stage just prior to larval hatching. However, the few individuals that did hatch completed development and reached adulthood (Supplementary Table 3). This points to genomic incompatibilities between nuclear loci that impact pre-hatching embryonic development. It remains unknown whether the F2 individuals that managed to reach adulthood were indeed balanced hybrids, or alternatively whether they derived mostly from one parental genotype. Hybrids between C2N2 and C3N3 displayed significantly reduced F2 egg hatching rates as well, but to a much lesser degree than crosses involving C1N1 (Supplementary Table 3). This higher hatching rate of C2N2 vs. C3N3 crosses agrees with the larger similarity between C2N2 and C3N3 genomes (Figs. 2, 4 and Supplementary Fig. 6).

Using these compatibility patterns as reference, we hypothesised that two lines would be compatible if their nuclear genomes were similar to each other, or that they would be incompatible if the differences between their nuclear genomes would be large. To test these hypotheses, we performed reciprocal crosses using several cyto-nuclear hybrid lines (Fig. 2, Supplementary Fig. 8). As expected, in incompatible crosses we found a significantly higher percentage of sterile F1 hybrid virgin females as compared to controls, and non-sterile F1 females laid fewer eggs than controls (Supplementary Table 4). Less than 3.5% of F2 eggs laid by these females hatched. Also, as expected, we found that in compatible crosses, F1 females were not more sterile than controls, laid as many or more eggs than controls, and more than 70% of their F2 eggs hatched (Supplementary Fig. 8, Supplementary Table 4). Thus, we related the genome-wide differentiation between C1N1 and either C2N2 or C3N3 to hybrid breakdown due to incompatibility at nuclear loci, as expressed most clearly in F2 recombinant individuals.

In conclusion, strong, yet incomplete post-zygotic barriers to gene flow exist between the honeysuckle race (C1N1) and co-occurring conspecifics with larger host ranges (C2N2 and C3N3). Barriers to gene flow expressed as F2 hybrid breakdown are not uncommon in haplodiploid taxa[81,82]. It is possible that the interactions between the nuclear and the mitochondrial genomes in hybrids contribute to the patterns in this system, as it is observed in haplodiploid taxa[79]. For example, although crosses between C2N2 and C3N3 were largely compatible, the fitness of F2 hybrids was asymmetrical: a higher proportion of individuals with a C2 cytotype hatched compared to the reciprocal hybrids with a C3 cytotype (49% vs. 23%, Fig. 4, Supplementary Table 3). Mito-nuclear interactions are found in populations that have recently experienced selection, as the mtDNA needs to accommodate the changes imposed by the selection regime acting on the nuclear genome in order to maintain homeostasis[79,83,84]. If embryonic development in *T. urticae* resembles larval development in *Nasonia* wasps and *Tigriopus* copepods, we might expect the oxidative phosphorylation pathway to delay or fully stop normal development of hybrid larvae due to an impaired ability to synthesize ATP[81,85].

**Fitness advantage of the honeysuckle race on its host over sympatric conspecifics**. To determine the extent of host adaptation in this system, we quantified fitness traits from the individual to the population levels. Due to its restriction to honeysuckle in the field, we expected C1N1 to either perform better on honeysuckle than on other host species tested, to outperform sympatric conspecifics with larger host ranges (i.e. generalists C2N2 and C3N3) on honeysuckle, or both[21,34].

We observed that reproductive performance (eggs/female/day) of individual C1N1 females (1–2 eggs/female/day) from several lines was more than twice lower than that of C2N2 (~5 eggs/female/day) and C3N3 females (~5 eggs/female/day) on leaf discs

of honeysuckle, a finding that was consistent as well for most other host plant species tested (Supplementary Fig. 9, Supplementary Table 5). Similar reproductive performance values were also obtained on detached honeysuckle twigs (C1N1 = $1.54 \pm 0.14$ [mean ± SEM]; C2N2 = $4.92 \pm 0.47$; $n = 12$). We focused on early individual female reproductive performance as this is an important trait determining the rate of population increase in spider mite populations[34,86,87]. The clutch sizes of C1N1 females were low relative to most reports of reproductive output in spider mite females[60]. Since the reproductive performance of C1N1 females is no better on honeysuckle than on other hosts tested, it is possible that genetic trade-offs in host adaptation are either low or absent, if host adaptation is determined by different loci on the hosts species we selected[34,87–91]. Alternatively, but not mutually exclusive, since C1N1 lines are largely inbred (Supplementary Fig. 5), the low reproductive performance of C1N1 females might indicate that they suffer from inbreeding depression. Spider mite populations can harbour genetic variation for the number of eggs laid per female, but low egg numbers may result from the expression of deleterious alleles in inbred individuals, rather than from trade-offs as a result of antagonistic pleiotropy of host-adaptation loci[58]. Furthermore, experimental inbreeding negatively affects the rate of female oviposition compared to outbred populations of *T. urticae*[92]. We argue that genetic load could still be prevalent in C1N1 females.

Despite the low reproductive performance of C1N1 females, they nonetheless build significantly larger populations on honeysuckle than conspecifics, as line C1N1a expanded from 30 to $643 \pm 243$ adult females after two generations, while line C2N3a only produced $188 \pm 80$ adult females ($F_{1, 10} = 6.88$, $p = 0.02$). We found that this is largely due to the higher survival of C1N1 juveniles compared to generalist juveniles on honeysuckle (Fig. 5). During the development from egg to adult (13–14 days for this species), the survival of juveniles of generalist lines dropped to ~0.8 around day 7, when most individuals were larvae, compared to C1N1 survival, which was close to 1 at that time. The survival of generalist juveniles kept decreasing significantly from that of C1N1 juveniles until day 11 in all experiments (Supplementary Table 5). Generalist juveniles mostly died trying to escape the leaf enclosures, as they got stuck on the wet enclosure barriers, or in sugary exudates on the leaf surface. By day 13, mortality in all lines increased due to adults trying to escape from the arenas. As a reference, survival of C1N1, C2N2 and C3N3 juveniles to adulthood was high (>90%) on common bean, a very permissive host to *T. urticae* (Supplementary Tables 2 and 3). This disparity in juvenile and maternal traits is not unusual[20]. In *T. urticae*, the genetic architecture determining juvenile survival seems to be decoupled from loci that determine female reproductive performance in some cases, but in other cases both traits increase together after experimental isolation to one host e.g. refs. [42,58,91,92].

Together, our data indicate that differences in fitness traits on honeysuckle exist between the honeysuckle race and the generalist conspecifics it co-exists with in nature. The honeysuckle race builds larger populations on its host than generalist conspecifics. This is, unintuitively, not due to a higher reproductive output of host race females over generalist conspecifics on honeysuckle, but rather due to a higher mortality of generalist juveniles on this host. This would give the host race a competitive advantage over conspecifics on honeysuckle. In contrast, the host race is possibly outcompeted on alternative hosts due to its low reproductive output (Supplementary Fig. 9, Supplementary Table 6), especially if the survival of generalist juveniles on non-honeysuckle hosts would be higher than on honeysuckle itself (e.g. Supplementary Table 2). Given the high mortality of generalists on honeysuckle, we could expect selection to act upon variation in mechanisms

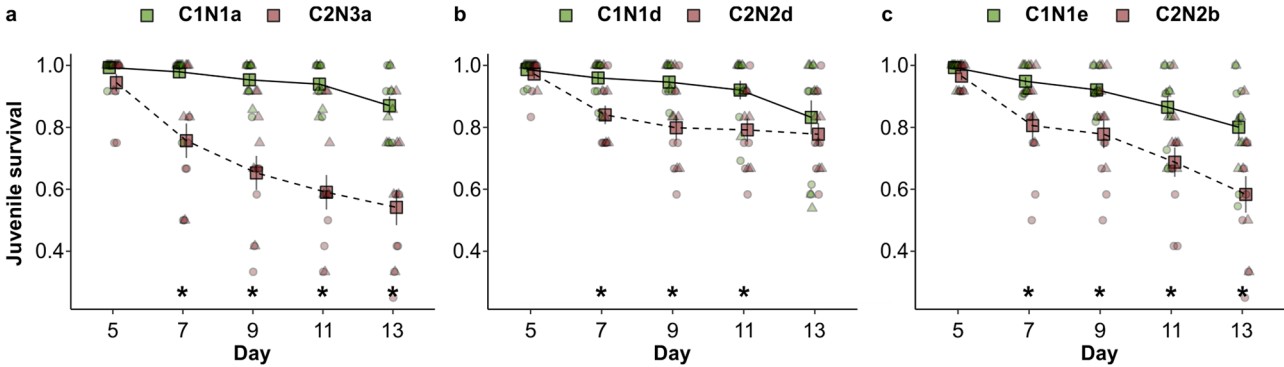

**Fig. 5 Proportion of juvenile survival of *Tetranychus urticae* field-derived lines on honeysuckle.** The survival proportion of juveniles (*y*-axes) from six iso-female lines representing common *T. urticae* genotypes found in the Dutch dunes was tested in three independent experiments (**a–c**). Honeysuckle-restricted lines C1N1a, C1N1d and C1N1e are coloured green and represented by a solid line; generalist lines C2N3a, C2N2d and C2N2b are coloured red and represented by a dashed line. Survival was scored across five consecutive time points, spanning from egg hatching to adulthood (represented in days on the x-axes). Stars represent a significantly different survival mean for each time point ($\alpha \leq 0.05$); error bars represent one SEM. $n = 12$ leaves for each genotype per panel, 6 from the top side of the leave (triangles) and 6 from the bottom side of the leave (circles). F-statistics and p-values are specified in Supplementary Table 5.

that prevent generalists from visiting this host, which would further restrict the opportunities for mating encounters and subsequent gene flow between honeysuckle specialists and generalist conspecifics, a possibility that remains to be tested empirically.

**Conclusions**. We present evidence of intraspecific population structure related to host use and incomplete reproductive isolation in the generalist herbivore *T. urticae*. Integrating lines of evidence of this process across scales (ecosystem–population–individual) proved imperative to grasp the extent of evolutionary divergence within this species in nature. The distribution of genetic variation of *T. urticae* across the Dutch dune ecosystem shows a consistent pattern across years, where honeysuckle-restricted genotypes co-exist with generalist conspecifics. Genetic structure was apparent genome-wide between the honeysuckle-restricted genotypes and other available genomes of *T. urticae*. The persistence of a genetically distinct, inbred population on honeysuckle in sympatry with conspecifics with larger host ranges is potentially due to strong but incomplete post-zygotic reproductive barriers between populations. Selection exerted by honeysuckle impacts negatively the survival of generalists on this host, which is expected to further reduce chances for hybridization with the host race. In nature, patterns of functional variation related to host use can be revealed using common genetic markers, but this is dependent on the choice of markers and on extensive genotyping of individuals within and between populations. Hybridization can occur in this system, but reproductive barriers are incomplete and expressed post-zygotically. Previous genetic studies with *T. urticae*, often with reproductively compatible green forms collected in greenhouses or agricultural settings, might suggest that host race formation in this generalist species is rare. Yet, host race formation could be more prominent in nature than in agricultural settings, where monocultures and pesticide use may erode genetic variation. Further research focusing on dissecting the patterns of intraspecific functional variation within herbivore taxa is needed to answer this question.

## Methods
**Associations between sympatric mite cytotypes and their host species in nature**. To investigate the occurrence of host-associated populations across spatial and temporal scales, populations of *T. urticae* were sampled in two coastal nature reserves near Castricum (CAS) and Meijendel (MEY) in The Netherlands, which

are part of the European dune ecosystem (Supplementary Table 1). Three to four ~1 km long transects were established within the forested dune areas of each nature reserve. Along a transect, 10 m × 10 m sampling locations were established at least 50 metres away from each other (CAS: 30 locations in total; MEY: 18 locations in total). Sampling took place in 2015, 2016 and 2017. At every location, multiple leaves from every host plant species showing signs of spider mite damage were collected and placed in plastic bags unique to location and host plant species to avoid cross-contamination. Within 48 h after field sampling, DNA was extracted from up to 10 individual mites, preferably adult females, per collection (Supplementary Note 1).

Individuals were genotyped by analysing SNPs in a 598 base pair (bp) stretch within the mitochondrial *cytochrome oxidase subunit 1* (*CO1*) gene; the so-called Folmer fragment[93]. The region was amplified using PCR and Sanger-sequenced (Supplementary Note 1). Sequence files were analysed and aligned using CodonCode Aligner (v. 6.0.2). To obtain a high-quality sequence, the ends of each sequence were trimmed to a final length of 598 bp. The haplotypes found have been deposited onto GenBank under project SUB7841957. All sequences were run through MegaBLAST to confirm their similarity to previously annotated *T. urticae* sequences. Sequences were aligned using the MUSCLE algorithm with a maximum of 5 iterations and a UPGMA clustering method in MEGA for Mac (v. 7). A maximum likelihood tree based on the Tamura-Nei nucleotide substitution model and bootstrapped 1000 times was constructed using all the unique haplotypes found across all samples. Clades with bootstrap support of 60% or above were used to define haplotype groups, which we referred to as cytotypes. Based on these clades, each individual mite was assigned to a particular cytotype. Host-mite associations were determined based on the proportion of samples that belonged to each cytotype that were sampled from a particular host plant species, per year. We analysed a subset of these data with samples from locations where we marked the co-occurrence of two wide-spread host species, wild honeysuckle (*Lonicera periclymenum*) and spindle tree (*Euonymus europaeus*), whether they were infested by mites or not. For cytotypes with evidence of host associations, we compared the proportion of individuals occurring on the alternative host to zero with a one sample *t*-test.

### Analysis of genomic divergence of honeysuckle-restricted lines relative to sympatric and non-sympatric conspecifics
*Establishment of lab populations, DNA extraction and endosymbiont diagnosis*. To quantify the extent of genetic divergence between host-associated populations, we established field-derived lines by re-sampling honeysuckle and spindle tree leaves at specific locations in CAS and MEY in August–September 2015 (Supplementary Table 1); infested leaves were collected as mentioned previously. Virgin females at the last moulting stage before adulthood (teleiochrysalis) were isolated individually on common bean (*Phaseolus vulgaris* cv. Speedy) leaf clippings of 3 × 3 cm surrounded by wet cotton wool, and kept under controlled conditions (25 °C, 16 h:8 h light:dark, 60% relative humidity; hereafter 'standard conditions'). Virgin females were allowed to oviposit for 7–10 days; their eggs developed under the same conditions, but the females were transferred to a colder chamber (12 °C, standard light regime and humidity) for ~10 days to slow down the aging process while their unfertilized eggs matured into adult males. Once the males emerged, the mother and her sons were placed together to mate on new bean cut-outs. After ~5 days of oviposition, females were collected for individual DNA extraction, *CO1* amplification and Sanger sequencing as described above. A total of 26 iso-female lines were established to represent the most common mite cytotypes previously identified and were maintained since their establishment on detached bean leaves in

standard conditions. After 2 weeks (~1 generation for this species), up to 800 adult females were pooled per line and their DNA extracted (Supplementary Note 2). The 26 pairs of reads generated as part of this project have been deposited onto sequence-read archive (SRA) under accessions SAMN13693727-52. Dilutions of 1:10 of this DNA and of DNA obtained from three laboratory populations known to harbour reproductive endosymbionts were used to diagnose the presence of bacterial symbionts by sequencing the V3 region of the 16S rDNA subunit on an Illumina MiSeq platform (Supplementary Note 3). The reads generated from the 29 bacterial communities have been deposited under accessions SRR12491964-89 and SRR12492212-14. Protocols followed for processing, mapping and variant calling, validation of species identity using nuclear sequences, quality control of the predicted variants, and assessment of heterozygosity levels in the field-derived lines are specified in Supplementary Notes 3–7.

*Mitochondrial genome and phylogenetic tree.* Illumina reads were mapped to the *T. urticae* mitochondrial genome using BWA (v. 0.7.12-r1039)[94]. We extracted the 13 protein-coding genes from the mitochondrial genomes for use in phylogenetic analyses, and individual alignments for each region were performed using MAFFT v. 7.721[95] with the following settings: strategy: auto; gap open penalty = 1.53, offset value = 0.0, and then visually IGV v. 2.3[96]. The 13 *T. kanzawai* mitochondrial genes were included in the analysis as an outgroup (Supplementary Note 4)[97]. Subsequently, the 13 regions were concatenated into a supermatrix containing 10225 bp. The best-fitting partitioning scheme was estimated using PartitionFinder2[98]. Several partitioning schemes were tested including division of protein-coding genes into 1st, 2nd and 3rd codon positions. The most suitable partitioning scheme was selected using the Akaike information criterion. Maximum likelihood (ML) phylogenetic analysis of the partitioned dataset was performed using RAxML (v. 8.2.8)[99], with BSbrL enabled and 500 bootstrap replicates to evaluate branch support.

*Nuclear phylogenetic tree.* A phylogenetic tree was constructed based on nuclear SNPs of all the iso-female lines from this study, as well as a previously published line of *T. kanzawai*[97] which served as an outgroup (Supplementary Note 4). To minimize the number of SNPs that were in linkage disequilibrium with each other, SNPs that were used for making the tree had to be spaced at least 20 kb apart. In the event of heterozygosity in a line at a given locus, the allele with higher read depth support was selected. From the resulting set of SNPs, a BioNJ[100] neighbour-joining tree based on observed nucleotide differences was constructed in PhyML 3.1[101] using the Seaview v. 4.6.1 graphical user interface[102] with 500 bootstraps.

*PCA and Fst estimation.* Principal component analyses (PCAs) and an Fst estimation at the nuclear genome level were performed in R (v. 3.3.2) as described by Zheng[103], using packages SNPRelate (v. 1.8.0) and gdsfmt version 1.10.1[104]. The PC analysis was performed by running the snpgdsPCA function with option autosome.only = FALSE. The snpgdsFst function was used to calculate Fst values following Cockerham and Weir[105]; autosome.only = FALSE. Additionally, Fst values were also calculated in a sliding window approach. Before conducting the sliding window analysis along the genome, the VCF file was transformed to be concordant with the recently reported three-chromosome assembly[47]. The R-package PopGenome[106] was used to perform pairwise Fst estimations for each window. Fst estimations were calculated based on the bi-allellic positions for windows of 10 kb and a window slide of 2 kb. The curves were smoothed by using the spline interpolation option (span = 0.01).

## Reproductive isolation between sympatric mite genotypes and genetics of cyto-nuclear hybrids

*Crossing bioassays between field-derived lines.* To quantify the extent of reproductive isolation between host-associated mite genotypes, reciprocal crosses between C1N1 and C2N2, C1N1 and C3N3, and C2N2 with C3N3 were carried out with individuals from several lines per genotype group. We quantified multiple fitness proxies in the parental crosses (P0) and in the first and second filial generations (F1 and F2) of the hybrid offspring, using intra-line crosses as controls in each experiment. We measured the sterility of P0 mated and F1 virgin females (frequency of females that did not lay any eggs), mated P0 and virgin F1 egg clutch size (the number of eggs) per female per day, P0 and F1 female mortality (average number of days alive during the experiments), F1 and F2 egg hatchability (percentage of viable eggs over the total number of eggs), F1 and F2 juvenile survival to adulthood (percentage of adult offspring over the number of viable eggs), and the sex ratio of F1 adult progeny (percentage of females over total number of adult offspring). In P0 to F1 experiments, 8–10 virgin females collected at the last juvenile stage before adulthood from an iso-female line were placed together with 4–5 adult males from either the same line for control, or males from a line belonging to a different genotype group for treatments, on a detached leaf disc (∅ = 24 mm) from common bean (*P. vulgaris* cv. Speedy). Forty-eight hours later, mated females were transferred to experimental boxes, where each female was placed individually on a new bean leaf disc (∅ = 15 mm) surrounded by wet cotton wool. For F1 to F2 experiments, F1 individuals were collected from parental crosses similar to the ones described above. Eight to ten female F1 virgin females were collected and placed on new discs for 48 hours without males. All experimental females were allowed to lay eggs for 24 h and were then transferred to a new leaf

disc; females were allowed to oviposit for a maximum of 5 days. A total of 951 females were analysed between all experiments, with an average of 50 females per crossing arrangement (Supplementary Tables 2 and 3).

## Measurement of fitness traits in the honeysuckle race on this host against sympatric conspecifics

*Reproductive performance on different host plants.* To gather evidence of host adaptation, we quantified several fitness proxies of the iso-female lines on different hosts. Oviposition of adult, mated females was measured as the number of eggs laid per mite, per day on detached leaf discs, for five consecutive days (eggs/female/day). An independent experiment was conducted for each of the five plant-species tested: common bean (*P. vulgaris* cv. Speedy), nightshade (*Solanum nigrum*), spindle tree (*E. europaeus*), stinging nettle (*Urtica* sp.), and honeysuckle (*L. periclymenum*). Bean and nightshade were grown under greenhouse conditions (25 °C, 16 h L:8 h D, 60%RH), spindle tree and stinging nettle were collected from the field, and honeysuckle was bought from a commercial supplier, defoliated and re-grown under greenhouse conditions. Before each experiment, virgin females in the last juvenile stage (teleiochrysalis) were selected from several laboratory lines belonging to C1N1, C2N2 or C3N3, and were placed together with adult males from the same line on bean leaf discs (∅ = 24 mm) for 48 h. Then, each ±1-day-old mated female was transferred to an individual leaf disc (∅ = 15 mm) of the experimental plant, surrounded by wet cotton wool and kept under standard conditions. Females were transferred to freshly cut leaf discs after 24 h, for a maximum of 5 days. Between 64 and 80 females from C1N1, C2N2 or C3N3 were tested on each plant species.

*Juvenile survival on detached honeysuckle twigs.* To infer fitness advantages of the honeysuckle host race over other genotypes on its host, we quantified the proportion of juveniles that survived to adulthood on honeysuckle. Three independent experiments were conducted, each using a different pair of lines: C1N1a and C2N3a; C1N1d and C2N2d; C1N1e and C2N2b. Experiments were conducted on detached honeysuckle twigs, consisting of two opposing leaves pressed against a bed of wet cotton wool, one with the top and one with the bottom side up. An area of approximately 2 × 3 cm within each leaf was delimited using thin paper tissue strips embedded into the wet cotton bed. An a priori limitation of the mite lines used here is that C2 females lay on average 5–6 eggs per day, whereas females from C1 females lay 1–2 eggs per day on honeysuckle and other hosts (see *Results*). Taking this into account, 16 adult females from C1 and 8 females from C2 per experiment were placed on each leaf of the experimental arenas. After 24 and 48 h, all eggs were counted and removed from the leaf surface, and any dead females were counted and replaced. Twenty-four hours later, all females were removed; eggs were removed until a total of 12 eggs were left on each leaf. A total of 12 leaves (6 with the top side up and 6 with the bottom side up) with 12 eggs each were used per line. The experiments spanned for the turnover of one generation for these lines (approx. 13 days under lab conditions). Survival scoring started from 5 days after egg laying and continued every 2 days for a total of five time points (day 5, 7, 9, 11 and 13), when over 50% of surviving individuals had reached adulthood. At each time point, the number of alive individuals and their life stage were scored, along with the cause of death of the dead individuals.

*Population growth on full honeysuckle plants.* Store-bought honeysuckle plants were defoliated, replanted in new soil and placed inside a mite-proof cage under standard greenhouse conditions. Experimental plants were used within 4 weeks of planting, but several replicates showed signs of infection by powdery mildew. The number of green leaves and estimates of the percentage of the plant covered by mildew were noted before mite infestation (T0). At T0, each plant was infested with 30 mites from population C1N1a or mites from population C2N3a. After two generations (i.e. ~40 days later), adult female mites were counted non-invasively, and damage by mildew was assessed as at T0. Two blocks of the experiment were performed consecutively and were set up with 4 replicates per line each, for a total of 8 replicates for C1N1a and 8 replicates for C2N3a. Plants were watered once a week during the experiments.

## Statistics and reproducibility

*Crossing bioassays between host-associated populations.* A mixed-effects linear model (package lme4 in R v. 3.6.1) was fitted to analyze the effect of hybridization on (1) P0 and F1 egg clutch size per female per day, (2) F1 and F2 mortality, (3) F1 and F2 egg hatchability, (4) F1 and F2 juvenile survival to adulthood, and (5) F1 sex ratio. 'Crossing treatment' was used as a fixed effect; iso-female line, and experimental box within experiment were used as independent random effects in each model. Sample sizes are specified in Supplementary Tables 2 and 3. When a main significant effect of the crossing treatment was found, significance of pairwise comparisons between crossing scheme pairs were determined using linear hypotheses with a Tukey correction for multiple testing. Basic model assumptions of residual normal distribution and homoscedasticity were assessed for each fitted model and data were log+1 transformed when these assumptions were violated. To compare the frequency of sterile versus fertile females between crosses, we performed a chi-square test of independence and a post-hoc test with multiple pairwise comparisons with a Bonferroni correction to account for multiple

comparisons. Sample sizes for crosses between compatible and incompatible lines are specified in Supplementary Fig. 8.

*Reproductive performance on different hosts.* Differences in the average eggs/female/day between nucleotypes were assessed using a linear mixed model in R (package *lme4* in R v. 3.6.1), in which the response variable was the number of eggs/female/day log+1 transformed to meet model assumptions. The average number of eggs/female/day was compared within each host tested, with nucleotype as the fixed factor with three levels (C1N1, C2N2 and C3N3) and iso-female line as the random factor. Between 64 to 80 females from C1N1, C2N2 or C3N3 were tested on each plant species. Females that died before day 1 of the experiment were eliminated from the analyses.

*Juvenile survival on detached honeysuckle twigs.* Differences in the mean proportion of surviving individuals per time point between members of each pair were analysed using a mixed-linear effects model (package *lme4* in R v. 3.6.1). As a response variable, we included the average proportion of individuals alive, square-root-transformed to meet model assumptions. The model parameters, fixed and random, were defined by comparing a full model to reduced versions of it and choosing the one with the lowest Akaike's information criterion (AIC) score. The full model was defined as: sqrt(alive.percentage) ~ genotype + (1| leaf side) + (1| genotype:leaf side) + (1| population:replicate) + (1| replicate:genotype). The best fitting model was a reduced model, defined as: sqrt(alive.percentage) ~ genotype + (1|leaf side) + (1| replicate:genotype), where mite iso-female line (genotype) was the fixed effect, and leaf side and replicate per genotype were set as random effects. Sample size $n = 12$ leaves, 6 with the bottom side up and 6 with the top side up.

*Population growth on full honeysuckle plants.* We fitted a mixed linear model to test whether the initial conditions of the experimental plants differed significantly between each other. A term representing the interaction between the number of green leaves in each cage with the percentage of mildew covering them (leaves * mildew) was used as the response variable, line was a fixed factor, and block was used as a random factor. Another mixed linear model was used to assess the differences in the number of adult females on the plants after ~40 days between lines. To meet model assumptions, the log-transformed number of females was used as a response variable, the interaction between the final number of green leaves found and their percentage of mildew, line, and number of days since the beginning of the experiment were fixed factors, and block was used as random factor. Sample size $n = 8$ populations of each mite line, separated in 2 experimental blocks of 4 replicates per mite line.

**Reporting summary**. Further information on research design is available in the Nature Research Reporting Summary linked to this article.

## Data availability

Sequence data that support the findings of this study have been deposited in the following databases: CO1 sequences in GenBank with the accession codes MT814055-MT814210; whole-genome sequences of field-derived mite lines in SRA with accessions SAMN13693727-52; 16S bacterial sequences of field-derived mite lines in SRA with accessions SRR12491964-89 and of laboratory lines with accessions SRR12492212-14; whole-genome sequencing of *Tetranychus urticae* green and red morphs, and *T. turkestani*, *T. kanzawai* sequences in SRA under bioproject PRJNA530192. Datasets used to generate Figs. 1, 4, 5, Supplementary Figs. 2, 8 and 9, and Supplementary Tables 2 and 3 are available in figshare[107]. Additional datasets that support the findings of this study are available from the corresponding author upon reasonable request.

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

## Acknowledgements

We thank Huyen Bui and Robert Greenhalgh for help with experiments or analyses and Astrid Bryon, Peter Demaeght, Sabina Bajda-Wybouw, Wannes Dermauw for help with inbreeding. This work was supported by the Institute for Biodiversity and Ecosystem Dynamics (IBED, TVL starters-grant), the Research Foundation Flanders (FWO, Belgium, G053815N to T.V.L.), the European Research Council (ERC) under the European Union's Horizon 2020 research and innovation program (ERC consolidator grant 772026- POLYADAPT to T.V.L. and 773902-SuperPests to T.V.L.) and the US National Science Foundation (award 1457346 to R.M.C.).

## Author contributions

E.V.P., T.V.L. and J.B. designed the experimental set up. E.V.P. conducted field surveys and experimental assays. S.S. and A.K. conducted the genomic analyses, under supervision of R.M.C. E.V.P., S.S. and A.K. analysed the data and drafted the manuscript. R.M.C., T.V.L. and J.B. supervised the writing, critically revised the manuscript and contributed to its final version. E.V.P., S.S. and A.K. designed the figures.

## Competing interests

The authors declare no competing interests.
