## [Peer Review File · Communications Biology]

Reviewers' comments:

Reviewer #1 (Remarks to the Author):

This paper uses multiple lines of evidence (field survey, genetics, reproductive isolation) to illustrate of a host race in a spider mite species. Though I like the implementation of multiple disciplinary approaches to describe lineages, I don't think that, in its current form, the paper is able to clearly demonstrate that 1) a host race is formed, or 2) it is the adaptation to host species that drives the genetic divergence.

Firstly, the authors give neither a clear definition of nor standards to test a host race, though this is necessary for the purpose of the paper. Host races are generally considered to be an intermediate stage of divergence (somewhere in the middle of zero divergence and completely different species) and often exhibit some degree of gene differentiation with limited amount of gene exchanges. However, in this paper, according to the genetic and genome-wide SNPs data, the "host race" (spider mites that feed on honeysuckles) is genetically very different from the rest of the species ($F_{st} = 0.46 \sim 0.54$) with no evidence of gene flow. In this case, why call this population a host race rather than a cryptic species? Especially when the hybrid fitness data also supports that this is a cryptic species that has a high degree of genetic incompatibility with the rest of the species.

Secondly, the performance experiment on different host species cannot prove that it is the host adaptation that drives the evolution of the "host race" or maintains the "host race" to coexist with other spider mite populations. The female fecundity of the "host race" on its native host plant is significantly lower than other genotypes, which is the opposite of local adaptation. Also, the authors showed that the "host race" (genotype N1) has much higher population growth rate on honeysuckles than other genotypes (N2, N3), which is consistent with local adaptation to host plant species. However, this pattern could be due to a flaw in the experimental design which the authors mentioned themselves: "most N2-N3 juveniles died due to trying to escape the leaf enclosure, where they [got] stuck on the wet enclosure barriers..." (line 378-379). If anything, this could imply strong host preference across different genotypes, which could be the evidence of adaptation to host plants. However, the authors fail to argue along this line of reasoning and instead try to argue based on population growth rates. I don't think there is solid evidence to suggest a true difference in population growth rates, so this is not a convincing explanation for enhanced performance of spider mite "host race" -genotype N1 on its native host plants.

Other than the major conceptual issues I raised above, one major problem in this paper's presentation is the confusing interchangeable use of different names for the same genetic lineage: C1, N1, or C1N1, all seems to refer to the same "host race". This problem is spread throughout the text and makes it very difficult to follow. Additionally, there are several grammatical errors throughout the paper.

I have some minor comments about the clarity of the text below:

Line 44: "...identify functional variants in nature". Not sure what functional variants is referring to here.

Line 50: The statement that herbivore arthropods are more speciose than non-herbivore arthropods is not necessarily true; parasitoids are probably the most speciose group. Check out this paper: Forbes et al. 2018 (<https://bmcecol.biomedcentral.com/articles/10.1186/s12898-018-0176-x>).

Line 83: need to define what "extreme generalists" is.

Line 102: Unclear what "this" refers to. If it refers to the previous pattern "isolation by distance" and increasing populations densities, it doesn't make sense to me.

Line 110: hybrid unviability is not a commonly used term. It should be hybrid inviability.

Line 130: "between field-derived lines and available genomes of *T. urticae*", unclear what this means.

Line 156: please report percentage of sequence divergence here.

Line 208: the term red and green forms came out of nowhere. Please specify what they are.

Reviewer #2 (Remarks to the Author):

I had the chance to read and evaluate COMMSBIO-20-2952-T. The authors explore whether mite *Tetranychus urticae* form host races in nature. The authors show the existence of extensive genetic differentiation even though races co-occur in sympatry. Host races also show strong postzygotic isolation which indicates the possibility of advanced genetic differentiation. Overall, all the work in the manuscript has been carefully executed and explained. The rationale for the analyses is clear. I have a few relatively minor comments that, in my opinion, would improve the clarity of the message:

Figure 4 should show the true distribution of the results. The reproductive compatibility between different lineages is one of the cruxes of the manuscript and the current representation only shows the reader the mean values of the trait. I would argue that the variance and range of the observations are quite important here. For example the results of the second row could be driven by just a few exceptionally low-compatibility observations or by a true depression of the compatibility. A boxplot/violin plot would reveal the range. Same applies to Figure S9.

Figure 5. It is not clear, at least to me, how many observations were included in the x-axis of this figure. Was it really a continuous observation? From my reading of the methods, it seems like this was a discrete measurement and perhaps representing the confidence intervals with a bars instead of ribbons would be more appropriate.

Line 543. Could you please define teleiochrysalis?

Appendix 9. Where the random effects independent from each other or were they nested?

Appendix 9. When were the $\log(i+1)$ transformations required.

Reviewer #3 (Remarks to the Author):

The paper by Villacis-Perez and colleagues describes the finding of a host race of the generalist spider mite *T. urticae* that exists in sympatry with other *T. urticae* populations that seem more generalists. The authors bring convincing evidence to the existence of a post-zygotic genetic barrier between the host race and the generalist populations. Finally, the authors argue that the selection on the host race host (honeysuckle) is involved in maintaining the gene pool of the host race from mixing with the generalist species.

The paper is largely divided into four chapters. The first three chapters discuss the major finding of a host race population of *T. urticae* under natural conditions, which is unique as this species is considered to be an extreme generalist. The data brought in chapter 2 using whole genome SNPs analysis to estimate genetic relationship which indicate very little gene flow (by PCA and F_{st}) between the host race and generalist populations are highly convincing and the analyses well conducted. This is also followed by convincing crossing experiments (chapter 3) that quantitatively show very significantly that although F1 females are produced between the host race and generalist populations, they are sterile and hardly produce an F2 generation (indicating post zygotic genetic barrier).

So, beside some minor comments (below), I agree with the conclusions drawn from the first three chapters and find them well supported, conducted and written.

That being said, I must state that I am in less agreement with the conclusions and the way the fourth chapter was conducted. In other words, I don't think that the experiments done and the data obtained can support associating the performance of the generalist and host race populations on honeysuckle and other hosts with the maintenance of genetic isolation.

First, the authors do not provide us with data on the performance of the host race on other hosts, so we do not actually know if it is indeed a host race or a specialist as claimed (no preference data either). We only know that it can perform on honeysuckle and on bean and on bean its performance is nearly not different from the generalist populations (Table S2 and S3). What about other hosts? We do know that females of the host race lay fewer eggs but this is also true for honeysuckle so it can't serve as a comparison parameter for other host. To my opinion, question 1 raised in line 344 remains unanswered until the end.

Second, to my best understanding, all the experimental iso-female lines were maintained on common bean since their establishment. Moreover, performance assays were conducted by transferring the lines to the experimental host from common bean. To my humble opinion, this is not the way to conduct and compare performance assays between populations. Based on the spider mites literature and my own experience, there is a possibility that after few generations of acclimation on the experimental host, the generalist lines will perform as well as the host race on honeysuckle for example. I would have not made a big issue out of it if the field survey data wouldn't have shown the presence of large numbers of the generalist genotypes on honeysuckle. As spider mites are not great migrators, I would assume that they might spend few generations on the same host without migration. Therefore, the lower performance reported for the first generation and with a limited sample size also for the second generation, might not reflect real field conditions. In any case, we also do not know if the generalist species perform better on other hosts than honeysuckle, or if it is actually their best host in the studied system.

Third, the authors report the host race population to be nearly isogenic. Could it be that this population was simply established from very few individuals that just by some random catastrophic event, not related to their host plant, made them genetically isolated from other *T. urticae* populations? In this case, the genetic isolation might be simply maintained because there is no way to go back. The only thing that might still happen is a switch from post- to pre-zygotic isolation to avoid the cost of sterile mating.

I am not asking the authors to conduct further experiments but to discuss and rebuttal the points raised here, or if accepted, to change chapter four and other relevant parts accordingly (even the title).

Other comments:

1. lines 56-61 – As the species highlighted are specialists, it might be better to add also the *Myzus persicae* population adapted to tobacco. You can start for example from "Nikolakakis, N. N., Margaritopoulos, J. T., & Tsitsipis, J. A. (2003). Performance of *Myzus persicae* (Hemiptera: Aphididae) clones on different host-plants and their host preference. *Bulletin of Entomological Research*, 93(3), 235–242".
2. line 73 – It is better also to cite "Forister, M. L., Novotny, V., Panorska, A. K., Baje, L., Basset, Y., Butterill, P. T., ... Drozd, P. (2015). The global distribution of diet breadth in insect herbivores. *Proceedings of the National Academy of Sciences of the United States of America*, 112, 442-447".
3. line 91 – May the author draw a clear line between host races and cryptic species and change their wording along the manuscript accordingly.
4. line 137 - My personal opinion is that this specific manuscript can benefit from dividing the results and discussion parts.
5. lines 146-149 – These lines are redundant with what comes next in more details, I suggest to omit these lines.
6. line 165 – Please indicate here that significant amount or % of C2 and C3 individuals were also found on honeysuckle.
7. line 178 – My personal opinion is that "De Barro, P. J. (2005). Genetic structure of the whitefly *Bemisia tabaci* in the Asia-Pacific region revealed using microsatellite markers. *Molecular Ecology*, 14, 3695-3718" is one of the best papers on the topic.

8. lines 310-312 – Not easy to guess where your hypotheses are going. Better to explain that you were mostly interested in the C2N1 expected compatible and incompatible crosses.
9. line 322-323 – Why is this argument true?
10. line 575 – Could not find these data in Table S1.

Point-by-point reply to the reviewers

Reviewer #1

Comment 1: *Firstly, the authors give neither a clear definition of nor standards to test a host race, though this is necessary for the purpose of the paper. Host races are generally considered to be an intermediate stage of divergence (somewhere in the middle of zero divergence and completely different species) and often exhibit some degree of gene differentiation with limited amount of gene exchanges.*

Response: The definition of a host race to which we adhere to throughout the article is now explicitly stated in lines 54-60 and it reads:

‘Following Drès & Mallet (2002), host races are defined here as populations that i) are associated with their hosts across spatial and temporal scales; ii) show some extent of genetic differentiation from sympatric conspecifics; and iii) show incomplete reproductive isolation from sympatric conspecifics. In addition, host races may differ from conspecifics in traits associated with host plant adaptation, but this criterion is not an absolute requirement for host race formation (Futuyma & Peterson 1985; Dres & Mallet 2002; Funk 2012).’

Comment 2: *However, in this paper, according to the genetic and genome-wide SNPs data, the “host race” (spider mites that feed on honeysuckles) is genetically very different from the rest of the species ($F_{st} = 0.46 \sim 0.54$) with no evidence of gene flow.*

Response: The F_{st} values mentioned by the reviewer are likely to be inflated due to the homozygosity of the honeysuckle race. We have made this point clear and cited the seminal work of Charlesworth et al 1997 in line 244 to more explicitly highlight how homozygosity introduces a bias to comparisons of population structure based on F_{st} . Our data support this claim. Figure 3, particularly panel A, shows that genetic differentiation of the honeysuckle host race is minimal in the context of all the genomic data available for this mite species. In addition, our crossing assays show that reproductive isolation is incomplete between mite lineages.

Comment 3: *In this case, why call this population a host race rather than a cryptic species? Especially when the hybrid fitness data also supports that this is a cryptic species that has a high degree of genetic incompatibility with the rest of the species.*

Response: While F2 hybrid breakdown is strong, it is not absolute (Figure 4, bottom row). Genetic incompatibility is incomplete, as viable and fecund F1 individuals are created from crosses between the host race and sympatric conspecifics. Since a cryptic species would fall within the biological definition of a species (full reproductive isolation), we believe that this term does not apply to the populations we describe in this article.

Comment 4: *Secondly, the performance experiment on different host species cannot prove that it is the host adaptation that drives the evolution of the “host race” or maintains the “host race” to coexist with other spider mite populations.*

Response: We agree that the previous version of the manuscript was not worded precisely with respect to this point, and allowed the interpretation that host adaptation was the driver of the evolution of host races. We have corrected and clarified the wording in the text, particularly in Results and Discussion section 4, to prevent the inference of causality from our data. However, we do want to stress that significant differences in fitness traits on honeysuckle are present between mite lineages, and we discuss that these differences may play a role in the co-existence of different mite lineages at the small spatial scales we examine.

Comment 5: *The female fecundity of the “host race” on its native host plant is significantly lower than other genotypes, which is the opposite of local adaptation. Also, the authors showed that the “host race” (genotype N1) has much higher population growth rate on honeysuckles than other genotypes (N2, N3), which is consistent with local adaptation to host plant species. However, this pattern could be due to a flaw in the experimental design which the authors mentioned themselves: “most N2-N3 juveniles died due to trying to escape the leaf enclosure, where they [got] stuck on the wet enclosure barriers...” (line 378-379).*

Response: Indeed, while we did not initially expect female reproductive performance to show such patterns, a clear (and very interesting) disparity between female reproductive performance, juvenile survival to adulthood, and population size was found. The values of population sizes we present in the article are not based on enclosures, as they were determined on whole plants. Only the values for juvenile survival are based on the laboratory set up (an experimental necessity); however, the patterns are consistent across experiments from the individual to the population levels.

Comment 6: *If anything, this could imply strong host preference across different genotypes, which could be the evidence of adaptation to host plants. However, the authors fail to argue along this line of reasoning and instead try to argue based on population growth rates. I don't think there is solid evidence to suggest a true difference in population growth rates, so this is not a convincing explanation for enhanced performance of spider mite “host race” -genotype N1 on its native host plants.*

Response: The data presented for juvenile survival and for population size on honeysuckle could indeed include the possibility that individuals try to escape the plant in search of other hosts, and perish when doing so. Any adaptation mechanisms (such as host preference) that lead to the observed differences remain to be empirically studied. Our focus was to quantify fitness traits related to the genetic background of the mite lineages, and we present strong differences of juvenile survival and population sizes after a unit of time. We do not claim to have estimations of population growth rate, measured as the population rate of increase (R_m or R_0). We have changed the wording to 'population size' to avoid confusions, particularly in line 392. Our changes to the wording of Results and Discussion section 4 hopefully avoids inferring causality of host adaptation in the patterns observed.

Comment 7: *Other than the major conceptual issues I raised above, one major problem in this paper's presentation is the confusing interchangeable use of different names for the same genetic lineage: C1, N1, or C1N1, all seems to refer to the same “host race”. This problem is spread throughout the text and makes it very difficult to follow.*

Response: We have carefully considered the reviewer's comment and thought of different options for the nomenclature of the lineages that we describe in this article. However, we strongly believe that the nomenclature that we use here (Cx for cytotype, Nx for nucleotype and CxNx for both of them together; i.e., genotype) allows us to separate the datasets gathered for each of the different sections of the manuscript. While field data is gathered only from the cytotype data, the nuclear genomic data is separated from the mitochondrial data in the whole-genome analyses. Furthermore, we have found mite populations that belong to different cytotype and nucleotype groups, and this nomenclature helps explain this finding. *However*, we have adjusted the text in order to separate the different datasets (cytotype, nucleotype and genotype) and avoid using them interchangeably. We believe our changes now make it clearer for the reader while still capturing the complexities of our study and its findings.

Comment 8: Additionally, there are several grammatical errors throughout the paper.

Response: We have found and corrected several grammatical errors.

Comment 9: *Line 44: "...identify functional variants in nature". Not sure what functional variants is referring to here.*

Response: Changed the text in line 44: '... identify herbivore populations associated to different plant species in nature'

Comment 10: *Line 50: The statement that herbivore arthropods are more speciose than non-herbivore arthropods is not necessarily true; parasitoids are probably the most speciose group.*

Response: Changed text in line 50 to: 'Herbivorous arthropods are highly speciose taxa...'

Comment 11: *Line 83: need to define what "extreme generalists" is.*

Response: Line 87. Changed the text to: 'The ecological interactions between generalists and the plethora of hosts they use through time in the field are not well understood...'

Comment 12: *Line 102: Unclear what "this" refers to. If it refers to the previous pattern "isolation by distance " and increasing populations densities, it doesn't make sense to me.*

Response: To give clarity to what 'this' means, we changed the text in line 106 to: 'Instead, genetic differentiation has been found to decrease when populations are closer to each other and when population densities are large...'

Comment 13: *Line 110: hybrid unviability is not a commonly used term. It should be hybrid inviability.*

Response: Line 116: We have changed 'unviability' for 'inviability'

Comment 14: *Line 130: "between field-derived lines and available genomes of T. urticae", unclear what this means.*

Response: For clarity, changed the text in line 130 to: '*ii*) screening for genome-wide and localized patterns of sequence differentiation using field-derived iso-female lines and previously sequenced populations of *T. urticae*;...'

Comment 15: *Line 156: please report percentage of sequence divergence here.*

Response: There may have been a misunderstanding; the requested information, which shows relative sequence divergence, is provided in Figure S1 for all the cytotypes found in this study.

Comment 16: *Line 208: the term red and green forms came out of nowhere. Please specify what they are.*

Response: This was an oversight on our part and is now corrected. We added this text in line 94: 'Two colour morphs of *T. urticae* occur across its cosmopolitan distribution, the green and the red morphs (Auger et al 2013). The role...'

Reviewer #2:

Comment 1: *Figure 4 should show the true distribution of the results. The reproductive compatibility between different lineages is one of the cruxes of the manuscript and the current representation only shows the reader the mean values of the trait. I would argue that the variance and range of the observations are quite important here. For example the results of the second row could be driven by just a few exceptionally low-compatibility observations or by a true depression of the compatibility. A boxplot/violin plot would reveal the range. Same applies to Figure S9.*

Response: We have converted figures 4, S8 and S9 to jittered boxplots to allow the visualization of the range of the data, and the respective figure captions have been modified to reflect this change.

Comment 2: *Figure 5. It is not clear, at least to me, how many observations were included in the x-axis of this figure.*

Response: Added to line 595: 'A total of six leaves with eggs were used per line.'

Comment 3: *Was it really a continuous observation? From my reading of the methods, it seems like this was a discrete measurement and perhaps representing the confidence intervals with a bars instead of ribbons would be more appropriate.*

Response: Indeed it was a discrete measurement, and therefore we changed the figure to include bars representing error. The figure captions have been changed accordingly.

Comment 4: *Line 543. Could you please define teleiochrysalis?*

Response: Changed text in line 559 to: ‘...virgin females collected at the last juvenile stage before adulthood from an iso-female line were placed...’

Comment 5: *Appendix 9. Where the random effects independent from each other or were they nested? Appendix 9. When were the $\log(i+1)$ transformations required.*

Response: We corrected the text to explicitly indicate that random effects were independent from each other, and to indicate that $\log(i+1)$ transformations were used when model assumptions of residual normal distribution and homoscedasticity were violated.

Reviewer #3:

Comment 1: *I must state that I am in less agreement with the conclusions and the way the fourth chapter was conducted. In other words, I don't think that the experiments done and the data obtained can support associating the performance of the generalist and host race populations on honeysuckle and other hosts with the maintenance of genetic isolation.*

Response: We regret that our wording assigned causality to host adaptation and genetic diversity. This was not our intention. The objective of our study was to describe a system from nature, where we observed different genotypes that show high levels of reproductive isolation. We suggest that the specialist inbred genotype persists on honeysuckle because it can compete only on this host with the other genotypes. Because of this pattern, we suggest that host association or adaptation likely plays a role in the co-existence of distinct mite lineages in this ecosystem, given the close proximity of mite populations, and the lack of complete reproductive isolation. We have adjusted to text throughout chapter 4, starting with line 343, to make it clear that this is a possibility, and not a direct result of our experiments.

Comment 2: *First, the authors do not provide us with data on the performance of the host race on other hosts, so we do not actually know if it is indeed a host race or a specialist as claimed (no preference data either). We only know that it can perform on honeysuckle and on bean and on bean its performance is nearly not different from the generalist populations (Table S2 and S3). What about other hosts? We do know that females of the host race lay*

fewer eggs but this is also true for honeysuckle so it can't serve as a comparison parameter for other host. To my opinion, question 1 raised in line 344 remains unanswered until the end.

Response: Our data focuses on the interaction between the mite lineages and honeysuckle, since it is the only host where the host race (genotype C1N1) is found, and also because it was one of the hosts where the two generalist genotypes were found. We did explore whether female reproductive performance differed on multiple hosts (Figure S9) to gain more insight into the associations of the different mite populations on the hosts found in the Dutch ecosystem (Figure 1). We have adjusted the text in line 344 to read: '...expected C1N1 to either perform better on honeysuckle than on other host species tested, to outperform sympatric conspecifics with larger host ranges (i.e., generalists C2N2 and C3N3) on honeysuckle, or both...'. These are now stated as expectations, not as questions. Then, we use our data to understand which expectations are met and which ones are not, and discuss the possible factors contributing to why these expectations were met or not. We have changed the text in line 391-393 to: 'Together, our data indicate that differences in fitness traits on honeysuckle exist between the honeysuckle race and the generalist conspecifics it co-exists with in nature. The honey-suckle race builds larger populations on its host than generalist conspecifics...'

Comment 3: *Second, to my best understanding, all the experimental iso-female lines were maintained on common bean since their establishment. Moreover, performance assays were conducted by transferring the lines to the experimental host from common bean. To my humble opinion, this is not the way to conduct and compare performance assays between populations.*

Response: To some extent, we agree with the reviewer. A perfect transplantation experiment in which the populations would have been brought from the field and split into experimental hosts would have been the most informative procedure. However, two factors prevented us from doing this. Firstly, a steady supply of plants would have been needed to keep the field populations in the laboratory and to perform experiments. This was not possible because honeysuckle and spindle tree are very difficult to grow in the laboratory due to their seasonality and environmental requirements. Secondly, we derived iso-female lines from the field populations, in order to fix some of the genetic variation found within these populations. This procedure is also necessary to obtain good genomic data that we could interpret. Therefore, we decided to perform our experiments by keeping mites in a common environment (i.e. bean) and then transferring them to the experimental hosts. This is also because we wanted to understand whether the genetic variation that was fixed when creating the iso-female lines was correlated to fitness traits on hosts found in nature. And we do see a strong correlation between genetic background and fitness on a host species.

Comment 4: *Based on the spider mites literature and my own experience, there is a possibility that after few generations of acclimation on the experimental host, the generalist lines will perform as well as the host race on honeysuckle for example. I would have not made a big issue out of it if the field survey data wouldn't have shown the presence of large numbers of the generalist genotypes on honeysuckle.*

Response: As the reviewer pointed out, generalist genotypes also occur on honeysuckle. Several of the generalist iso-female lines used in our study were collected from honeysuckle.

In the field, generalists co-occur at very small spatial scales with the honeysuckle race. If acclimatization to honeysuckle would cause the generalist to perform as well or better than the honeysuckle race, then the honeysuckle race would most likely become extinct due to being outcompeted by the generalist in the field, which lays larger egg clutches than the host race. Furthermore, because we created iso-female lines and kept them in a common garden, we can infer that the fitness differences observed in our experiments are related to the mite genetic background. Lastly, genetic variation in the generalist lines is high, so there is a possibility that variants within the generalist populations would be able to adapt to honeysuckle too. This remains to be studied empirically.

Comment 5: *As spider mites are not great migrators, I would assume that they might spend few generations on the same host without migration. Therefore, the lower performance reported for the first generation and with a limited sample size also for the second generation, might not reflect real field conditions.*

Response: We were interested in understanding the correlation between genetic background and fitness on a host, and therefore we fixed the genetic variation of the mite populations found in the field by creating multiple iso-female lines belonging to each mite lineage. These lines took several generations to grow into populations we could use for experiments. Furthermore, it is likely that the low female reproductive performance of the honeysuckle race is related to their high levels of homozygosity.

Comment 6: *In any case, we also do not know if the generalist species perform better on other hosts than honeysuckle, or if it is actually their best host in the studied system.*

Response: The limited number of hosts tested in our experiments may indeed do not show the full breadth of hosts that these mite lineages can exploit, but we investigated female reproductive performance on several hosts that occur in the dune ecosystem (Figure S9). However, the point of this study is to understand whether mite genetic background is related to host use, and therefore our experiments are focused on the host where all mite lineages were found: honeysuckle. On honeysuckle, the generalist clearly performs worse than the honeysuckle race in terms of juvenile survival to adulthood and population sizes.

Comment 7: *Third, the authors report the host race population to be nearly isogenic. Could it be that this population was simply established from very few individuals that just by some random catastrophic event, not related to their host plant, made them genetically isolated from other *T. urticae* populations? In this case, the genetic isolation might be simply maintained because there is no way to go back. The only thing that might still happen is a switch from post- to pre-zygotic isolation to avoid the cost of sterile mating.*

Response: Indeed, in the field, we do not know if the honeysuckle race was established from a few individuals that went through a bottleneck unrelated to host adaptation; however, it is not the objective of this study to understand the origins of the honeysuckle race, but to describe the system that we observe in nature. Having identified the homozygosity of the honeysuckle race is one of the major results of this study. Importantly, host races may be formed without experiencing disruptive selective pressure from their hosts, and rather be formed by the evolution of reproductive isolation due to innate genetic incompatibilities.

Therefore, even if such a catastrophic event could happen in nature, it is independent of the idea that host race formation has taken place in this system.

Comment 8: *I am not asking the authors to conduct further experiments but to discuss and rebuttal the points raised here, or if accepted, to change chapter four and other relevant parts accordingly (even the title).*

Response: We have adjusted the text in chapter four and the title of our article to accommodate the comments brought up by the reviewer. Importantly, we changed the text in line 342 to set the expectations of the chapter and in lines 391-393 to synthesize our findings.

Comment 9: *Lines 56-61 – As the species highlighted are specialists, it might be better to add also the *Myzus persicae* population adapted to tobacco. You can start for example from “Nikolakakis, N. N., Margaritopoulos, J. T., & Tsitsipis, J. A. (2003). Performance of *Myzus persicae* (Hemiptera: Aphididae) clones on different host-plants and their host preference. *Bulletin of Entomological Research*, 93(3), 235–242”.*

Response: Added to line 64: “...Peccoud et al. 2009) , in populations of the peach-potato aphid *Myzus persicae* infesting tobacco (Nikolakakis et al. 2003; Margaritopoulos et al. 2007), and among other herbivores...”

Comment 10: *Line 73 – It is better also to cite “Forister, M. L., Novotny, V., Panorska, A. K., Baje, L., Basset, Y., Butterill, P. T., ... Drozd, P. (2015). The global distribution of diet breadth in insect herbivores. *Proceedings of the National Academy of Sciences of the United States of America*, 112, 442-447”.*

Response: Added this reference in line 79

Comment 11: *Line 91 – May the author draw a clear line between host races and cryptic species and change their wording along the manuscript accordingly.*

Response: We have adjusted the text and deleted the term 'cryptic species' in lines 86, 89 and 98

Comment 12: *Line 137 - My personal opinion is that this specific manuscript can benefit from dividing the results and discussion parts.*

Response: We have carefully considered this suggestion, but have concluded that given the large amount of data we use to describe patterns in nature, the reader will benefit much more from getting context, data and interpretation in a single 'results and discussion' section, as opposed to separate sections.

Comment 13: *Lines 146-149 – These lines are redundant with what comes next in more details, I suggest to omit these lines.*

Response: We deleted this text

Comment 14: *Line 165 – Please indicate here that significant amount or % of C2 and C3 individuals were also found on honeysuckle.*

Response: Added to line 163: “...from zero, and a significant proportion of C2 and C3 individuals were also found on honeysuckle (Figure S2).”

Comment 15: *Line 178 – My personal opinion is that “De Barro, P. J. (2005). Genetic structure of the whitefly Bemisia tabaci in the Asia–Pacific region revealed using microsatellite markers. Molecular Ecology, 14, 3695-3718” is one of the best papers on the topic.*

Response: Added this reference in line 176

Comment 16: *Lines 310-312 – Not easy to guess where your hypotheses are going. Better to explain that you were mostly interested in the C2N1 expected compatible and incompatible crosses.*

Response: To make our hypotheses clearer in line 311 to we changed the text from: “To evaluate whether these patterns depended on the compatibility of nuclear genomes, we performed similar reciprocal crosses between several cyto-nuclear hybrid lines and lines we hypothesized to be either compatible or incompatible (Figures 2 and S8).” to: “Using these compatibility patterns as reference, we hypothesized that two lines would be compatible if their nuclear genomes were similar to each other, or that they would be incompatible if their nuclear genomes would differ significantly from each other. To test these hypotheses, we performed reciprocal crosses using several cyto-nuclear hybrid lines (Figures 2 and S8).”

Comment 17: *Line 322-323 – Why is this argument true?*

Response: We have changed the text to avoid overextending the interpretation of our results, and rather keep our work as a description of a pattern observed in the field. The text in line 324 was changed to: ‘In conclusion, strong, yet incomplete post-zygotic barriers to gene flow exist between the honeysuckle race (C1N1) and co-occurring conspecifics with larger host ranges (C2N2 and C3N3).’

Comment 18: *Line 575 – Could not find these data in Table S1.*

Response: We corrected this mistake in line 577 by changing the text to ‘Between 64 to 80 females from C1N1, C2N2 or C3N3 were tested on each plant species.’

REVIEWERS' COMMENTS:

Reviewer #1 (Remarks to the Author):

The authors did a great job at improving the manuscript and addressing my previous concerns. I have no further comments.

Reviewer #2 (Remarks to the Author):

This new version of the manuscript addressed all my concerns, and from my assessment, did a thorough job with the comments from all the other reviewers as well.